# Identifying GPSM Family Members as Potential Biomarkers in Breast Cancer: A Comprehensive Bioinformatics Analysis

**DOI:** 10.3390/biomedicines9091144

**Published:** 2021-09-03

**Authors:** Huy-Hoang Dang, Hoang Dang Khoa Ta, Truc T. T. Nguyen, Gangga Anuraga, Chih-Yang Wang, Kuen-Haur Lee, Nguyen Quoc Khanh Le

**Affiliations:** 1International Ph.D. Program for Cell Therapy and Regeneration Medicine, College of Medicine, Taipei Medical University, Taipei 11031, Taiwan; danghuyhoang@gmail.com; 2Ph.D. Program for Cancer Molecular Biology and Drug Discovery, College of Medical Science and Technology, Taipei Medical University and Academia Sinica, Taipei 11031, Taiwan; d621109004@tmu.edu.tw (H.D.K.T.); g.anuraga@unipasby.ac.id (G.A.); chihyang@tmu.edu.tw (C.-Y.W.); khlee@tmu.edu.tw (K.-H.L.); 3Graduate Institute of Cancer Biology and Drug Discovery, College of Medical Science and Technology, Taipei Medical University, Taipei 11031, Taiwan; 4Translational Medicine Division, Graduate Institute of Biomedical Informatics, Taipei Medical University, Taipei 110, Taiwan; m610109011@tmu.edu.tw; 5Memory and Dementia Unit, Hospital 30-4, Ho Chi Minh City 70000, Vietnam; 6Department of Statistics, Faculty of Science and Technology, Universitas PGRI Adi Buana, Surabaya, East Java 60234, Indonesia; 7Cancer Center, Wan Fang Hospital, Taipei Medical University, Taipei 11031, Taiwan; 8Professional Master Program in Artificial Intelligence in Medicine, College of Medicine, Taipei Medical University, Taipei 106, Taiwan; 9Research Center for Artificial Intelligence in Medicine, Taipei Medical University, Taipei 106, Taiwan; 10Translational Imaging Research Center, Taipei Medical University Hospital, Taipei 110, Taiwan

**Keywords:** breast cancer, G-protein signaling modulator, survival analysis, biomarker, functional enrichment analysis, gene expression, immunotherapy

## Abstract

G-protein signaling modulators (GPSMs) are a class of proteins involved in the regulation of G protein-coupled receptors, the most abundant family of cell-surface receptors that are crucial in the development of various tumors, including breast cancer. This study aims to identify the potential therapeutic and prognostic roles of GPSMs in breast cancer. Oncomine and UALCAN databases were queried to determine GPSM expression levels in breast cancer tissues compared to normal samples. Survival analysis was conducted to reveal the prognostic significance of GPSMs in individuals with breast cancer. Functional enrichment analysis was performed using cBioPortal and MetaCore platforms. Finally, the association between GPSMs and immune infiltration cells in breast cancer was identified using the TIMER server. The experimental results then showed that all GPSM family members were significantly differentially expressed in breast cancer according to Oncomine and UALCAN data. Their expression levels were also associated with advanced tumor stages, and GPSM2 was found to be related to worse distant metastasis-free survival in patients with breast cancer. Functional enrichment analysis indicated that GPSMs were largely involved in cell division and cell cycle pathways. Finally, GPSM3 expression was correlated with the infiltration of several immune cells. Members of the GPSM class were differentially expressed in breast cancer. In conclusion, expression of GPSM2 was linked with worse distant metastasis-free outcomes, and hence could potentially serve as a prognostic biomarker. Furthermore, GPSM3 has potential to be a possible target for immunotherapy for breast cancer.

## 1. Introduction

Worldwide, breast cancer remains the most commonly seen malignancy in women, with a staggering incidence of 2 million cases annually [1]. Molecular markers of breast cancer, including estrogen and progesterone receptors and human epidermal growth factor 2, are crucial to the classification, treatment, and prognosis of individuals with this disease [2]. While early-stage, non-metastatic breast cancer is considered treatable, such curative therapies do not yet exist for patients in the more advanced stages with distant organ metastases [3]. Combinations of novel biomarkers play a critical role in the highly complex diagnostic and treatment algorithms of breast cancer [4]. Therefore, efforts to identify additional, robust biomarkers that could stratify patients and improve clinical outcomes are wholly justified.

In many instances, cancer arises from disruptions of cell signaling pathways, which modulate various cellular processes including survival, reproduction, and motility [5]. Among their components, G proteins, also known as guanine nucleotide-binding proteins, remain one of the most significant and are involved in the transduction of extracellular signals into the cell’s interior [6]. These intracellular molecular switches are triggered by the membrane-bound G protein-coupled receptors, and the ensuing signaling events eventually lead to changes in cell function. This process is further regulated by another group of proteins, G protein-signaling modulators (GPSM), via their interaction with subunits of G proteins. GPSMs are a class of receptor-independent activators of G protein signaling, whose family members include GPSM1, GPSM2, GPSM3, and GPSM4. In particular, GPSM2 is widely known to modulate mitotic spindle orientation [7], and its significance in breast cancer has been reported by earlier studies. Fukukawa et al. showed that GPSM2 is overexpressed in breast cancer tissues [8], while Deng et al. stated that its nuclear expression is an unfavorable prognostic indicator [9]. Furthermore, bioinformatics analysis and in vivo verification have suggested that GPSM2 negatively influenced patient response to paclitaxel, one of the most effective and well-tolerated chemotherapy drugs for breast cancer [10]. Meanwhile, members of the GPSM family are known to be associated with other neoplasms, namely lung [11], prostate [12], and pancreatic cancer [13].

A few reports to date have illustrated the expression profiles, as well as therapeutic and prognostic role of GPSM class in breast cancer. Thanks to an increasing number of publicly available databases and web tools for cancer-related genomics studies, researchers are able to leverage these resources to discover novel targets and biomarkers [14]. In this study, we aim to better understand the significance of GPSM family members as potential molecular therapeutic targets and biomarkers in breast cancer, through performing an integrated bioinformatics analysis. Our analytic workflow follows the standard bioinformatics studies [15,16] and it is briefly outlined in Figure 1. First, we determine whether GPSM family genes could serve as potential biomarkers by comparing their mRNA expression levels in breast cancer and normal samples, using Oncomine and UALCAN databases. After confirming that GPSMs were differentially expressed in breast cancer, their protein expressions in immunohistochemically stained breast cancer tissues were visualized by querying the Human Protein Atlas platform. Next, GPSMs’ connection with prognosis of patients with breast cancer was plotted using the Kaplan–Meier plotter. Univariate and multivariate Cox analyses were also performed to assess the proportional hazards model. Finally, we identify the GPSMs-related signaling pathways by overlapping datasets from METABRIC and TCGA, and explore the relationship between GPSMs and immune infiltration levels in breast cancer.

## 2. Materials and Methods

### 2.1. Oncomine

Oncomine (http://www.oncomine.org accessed on 15 June 2021) is an online DNA microarray database that allows cancer transcriptome analysis, and currently houses 715 datasets and 86,733 samples, covering the majority of cancer types [17]. It was used to compare the transcriptional levels of GPSM family members between tumor and normal samples in 20 types of cancer. Statistics were performed by Student’s *t*-test, and thresholds were set as follows: *p* value < 0.05, fold change >2, gene rank top 10%.

### 2.2. CCLE (Cancer Cell Line Encyclopedia)

The Broad Institute CCLE (http://www.broadinstitute.org/ccle accessed on 15 June 2021) is a dataset containing more than 1100 cell lines for 37 types of cancer [18]. It allows computational genomic analysis and visualization, and was used in our study to demonstrate the expression levels of GPSMs in breast cancer.

### 2.3. UALCAN 

UALCAN (http://ualcan.path.uab.edu accessed on 15 June 2021) is a web portal that allows online analysis of transcriptome sequencing data from The Cancer Genome Atlas (TCGA) project [19]. We used UALCAN to validate the relative expressions of GPSMs in normal and breast cancer samples, as well as those in different tumor stages and subclasses. This measurement was represented as transcript per million reads, and a comparison was carried out using Student’s *t*-test.

### 2.4. Human Protein Atlas

The Human Protein Atlas (https://www.proteinatlas.org/ accessed on 15 June 2021) contains mRNA and protein expression data of normal and cancerous human tissues [20]. mRNA and protein expression profiles of GPSMs in normal and breast cancer tissues were analyzed using RNA-sequencing data and immunohistochemical images.

### 2.5. Survival Analysis

The Kaplan–Meier plotter (https://kmplot.com/analysis accessed on 15 June 2021) is an online tool that assesses survival data of 54,000 genes in 21 cancer types, including breast, ovarian, lung, and gastric cancer [21]. The effect of GPSMs mRNA expression on distant metastasis-free survival (DFMS) of patients with breast cancer were identified by sorting patients into high and low median expression groups.

Next, univariate and multivariate Cox analyses were performed, using the *survival* package of R language (https://cran.r-project.org/web/packages/survival/index.html accessed on 15 June 2021), to assess the proportional hazards model. Variables including age, gender, tumor stage (TNM), and treatment.

### 2.6. Functional Enrichment Analysis

cBioPortal for Cancer Genomics (https://www.cbioportal.org/ accessed on 15 June 2021) is a portal for analyzing multidimensional gene-level data of cancer tissues and cell lines [22]. Two datasets from cBioPortal, namely METABRIC (Molecular Taxonomy of Breast Cancer International Consortium) and TCGA, were accessed to determine functional enrichment analysis [23]. In particular, we first identify the gene–gene interaction network using the Gene Multiple Association Network Integration Algorithm (GeneMANIA, https://genemania.org accessed on 15 June 2021) [24]. The Venny tool (https://bioinfogp.cnb.csic.es/tools/venny accessed on 15 June 2021) was then employed to find the overlap from lists of co-expressed genes from METABRIC and TCGA.

Next, MetaCore (https://portal.genego.com accessed on 15 June 2021), an integrated functional analysis software, was curated to determine the biological processes, biomarker networks, and the breast cancer cell–cell signaling pathway.

Finally, functional annotation was performed using GO and KEGG analyses. Gene ontology (GO) is the standardized representation of gene and gene products, covering three domains: Cellular component, molecular function, and biological process [25]. The Kyoto Encyclopedia of Genes and Genomes (KEGG) is the knowledge base for gene functions systemic analysis [26]. Results were visualized by R, package DOSE (Disease Ontology Semantic and Enrichment, version 3.13) [27].

### 2.7. TIMER (Tumor IMmune Estimation Resource)

TIMER (http://timer.cistrome.org/ accessed on 15 June 2021) is a web server for systematic analysis of immune infiltrates across various cancer types [28]. Its algorithm estimates the correlations between gene expression and an abundance of six immune infiltrates cells (B cells, CD4+ T cells, CD8+ T cells, neutrophils, macrophages, and dendritic cells). All Spearman’s correlation coefficients were tumor purity-adjusted.

## 3. Results

The basic characteristics, including their approved symbols, IDs, alternative names, and chromosomal locations, of all members of the GPSM family are shown in Table 1. It is worth noting that GPSM4 is more commonly known as PCP2 (Purkinje cell protein 2).

### 3.1. Differential Expression of GPSM Family Genes in Breast Cancer

Analysis from Oncomine revealed that all four GPSMs were differentially expressed in breast cancer (Figure 2A). In particular, GPSM1 and GPSM4 were downregulated while GPSM2 and GPSM3 were over-expressed. Detailed overexpression profiles of GPSM2 between cancer and normal tissues are provided in Appendix A.

Next, expression patterns of the GPSM family in various cell lines were analyzed from the CCLE database. The molecular subtypes of cell lines were also shown [29]. GPSM1, GPSM2, and GPSM3 were most highly expressed in triple negative breast cancer cell lines, while GPSM4 showed high expression in multiple luminal A cell lines (Figure 2B).

### 3.2. GPSMs Expression in Subgroups of Individuals with Breast Cancer

Regarding the comparison between healthy and breast cancer samples, UALCAN analysis illustrated higher transcription levels of all four GPSM genes in breast cancer patients (Figure 3A). For GPSM1, this result was not consistent with that from Oncomine and the Human Protein Atlas, which indicated that GPSM1 was under-expressed and had low protein expression status in individuals with breast cancer.

In terms of stratification by cancer stages, all GPSMs expression levels in healthy controls were significantly lower than those at stage 1, 2, and 3 (Figure 3B). GPSM3 and GPSM4 also showed similar patterns for stage 4 breast cancer. With respect to breast cancer subclasses, all GPSMs’ expression statuses were lower in normal individuals compared to those with luminal or triple negative breast cancer (Figure 3C). Detailed statistics of the comparison are provided in Appendix A.

### 3.3. GPSMs Expression Profiles on mRNA and Protein Level

Expression levels of GPSMs in non-cancer tissues were derived from the GTEx (Genotype-Tissue Expression) dataset in the Human Protein Atlas (Appendix A). Among all members of the GPSM family, GPSM3 and GPSM2 were most and least abundant in normal breast tissues (pTPM of 16.8 and 5.9, respectively). Furthermore, analysis revealed that GPSM1 had weak protein expression, in contrast to the moderate expression of GPSM2 in clinical breast cancer specimens (Figure 4). Corresponding data of GPSM3 and GPSM4 were not found in the database.

To summarize, in the GPSM family genes, GPSM2 expression was consistently higher in people with breast cancer than in normal controls at both mRNA and protein levels.

### 3.4. Prognostic Roles of GPSMs in Breast Cancer

The Kaplan–Meier plot was used to identify prognostic significance of the GPSM family gene in individuals with breast cancer. Among the four members, only GPSM2 is consistently associated with poor distant metastasis-free survival (DMFS) outcomes (Figure 5). In particular, higher expression levels of GPSM2 indicated worse DMFS time overall and in the following subtypes of breast cancer: ER or PR positive, HER2 positive or negative, luminal A or B, and with or without chemotherapy.

Meanwhile, survival results of the remaining members of GPSM family were more inconsistent (Appendix A). Specifically, while a higher level of GPSM1 mRNA showed significant improvement in outcome for patients with HER2+/− and luminal A and B subtypes of breast cancer, it was linked to poor prognosis in patients with ER+/−, PR +/−, and chemotherapy-included subtypes (S2-1). Furthermore, GPSM3 and GPSM4 with low expression levels were considerably associated with better survival outcomes (S2-2, S2-3).

As the Kaplan–Meier curve showed that higher GPSM2 expression and poor DMFS were consistently and significantly correlated, we next investigated whether GPSM2, along with variables such as age, gender, and tumor stage, were risk factors for survival in breast cancer patients. Univariate Cox analysis revealed that advanced age, TNM stage III/IV, and metastasis, but not GPSM2 expression, were independent high-risk factors for breast cancer survival. Subsequent multivariate Cox analysis showed that these three risk factors remained to be associated with decreased overall survival (Table 2).

### 3.5. Biological Functions and Pathway Enrichment Analysis

Figure 6 illustrates the gene interaction networks of GPSMs and their neighboring genes. The relationship between co-localization, shared protein domains, co-expression, prediction, and pathways of GPSMs is shown. Notably, we found that GPSM members were highly correlated with metastatic markers, such as ATG3 (autophagy-related gene 3) [30], DTX3 (Deltex E3 Ubiquitin Ligase 3) [31], and PALM [32]. Next, GO and KEGG resources were used to explore the functions of GPSM and their neighboring genes (Figure 7). We found that GPSMS were mostly involved in cell division from their “Biological Process” results: Nuclear division, organelle fission, chromosome segregation, mitotic nuclear division, and nuclear chromosome segregation. Subcellular locations of GPSMs were mostly around the chromosomal region, spindle, and chromosome and centromeric region. Molecular functions of these genes were mostly related to ATPase activity, catalytic activity (acting on DNA), and tubulin binding. KEGG pathways that were involved in breast cancer pathogenesis included cell cycle, oocyte meiosis, and cellular senescence. Finally, Disease Ontology analysis revealed that GPSMs played roles in autosomal dominant disease, non-small cell lung carcinoma, and hereditary breast ovarian cancer.

Next, we illustrate the functional pathway enrichment analysis from MetaCore, which involves determining gene networks and biological processes of each GPSM family member co-expressed genes. Again, among all GPSMs, GPSM2 is most relevant to breast cancer tumor biology (Figure 8). In particular, GPSM2 co-expressed genes play essential roles in the cell cycle and DNA damage pathways in breast cancer growth. The five pathways that showed the most significant *p* values are “cell cycle_role of APC (Anaphase-promoting complex) in cell cycle regulation”, “cell cycle_metaphase checkpoint”, “cell cycle_spindle assembly and chromosome separation”, “DNA damage_intra S-phase checkpoint”, and “cell cycle_start of DNA replication in early S phase”.

Similar analyses of GPSM1, GPSM3, and GPSM4 showed that their related biological pathways are only minimally linked to breast cancer development (Appendix A). For example, “prolactin/ERK (extracellular signal-regulated kinase) signaling in breast cancer” was ranked ninth among GPSM1 statistically significant pathway analysis, and “mitogenic action of ESR1 (Estrogen Receptor 1) (membrane) in breast cancer” was ranked 15th for GPSM4. Biological process analyses revealed that GPSM1, GPSM3, and GPSM4 showed biological significance in colon cells, adipose tissue, and neuronal cell, respectively.

### 3.6. Correlation between GPSMs and Tumor Purity and Immune Infiltrate in Breast Cancer

Knowledge regarding the host immune system is crucial to developing cancer immunotherapy [28], and high levels of tumor-infiltrating lymphocytes are associated with better prognosis [33]. TIMER analysis revealed that GPSM1, GPSM2, and GPSM4 were only slightly correlated with tumor purity and infiltrating lymphocytes (correlation coefficients ranging from approximately −0.3 to 0.2) (Figure 9). On the other hand, GPSM3 showed significant negative correlation with tumor purity (r = −0.48, *p* = 2 × 10^−58^), and moderate to strong positive association with six types of immune infiltrate cells (dendritic cell, r = 0.65; CD4+ T cell, 0.64; neutrophil, 0.56; B cell, 0.44; CD8+ T cell, 0.36; and macrophage, 0.29; all *p* values < 10^−20^). We further clarify the association between GPSM3 expression and immune infiltrates in different subtypes of breast cancer (Figure 10). The results demonstrated that GPSM3 expression was most strongly correlated with levels of CD8+ T cells, B cells, macrophage, and myeloid dendritic cells in subtypes basal and HER2 of breast cancer. Taken together, it is suggested that GPSM3 expression was related to immune infiltration levels in breast cancer.

## 4. Discussion

Breast cancer is the most prevalent malignancy in women globally [3]. It is widely acknowledged as a highly heterogeneous disease at multiple levels, from clinical and histopathologic classification to the expression of predictive and prognostic biomarkers, all of which must be taken into account to design targeted and personalized treatment plans for patients [34]. Considerable efforts are being directed at detecting accurate biomarkers as leads to anti-cancer drugs, or to predict treatment outcome and survival in breast cancer [35,36,37,38]. Among them, accumulating evidence suggests that signal transduction pathways, the process in which extracellular molecules interact with receptors to bring about changes in and around the cells, are highly attractive targets in breast cancer treatment and prevention, specifically by modulating G protein-coupled receptors [39,40,41]. A recent review by Usman et al. highlighted several ongoing anti-GPCR drugs for individuals with cancer [42]. Thus, further investigation of the roles of GPCRs and their modulators, including GPSMs, are warranted. To our knowledge, this is the first study that evaluates the potential therapeutic and prognostic value of all members of the GPSM family gene together, and their roles in breast cancer specifically. Prior studies have confirmed the contributions of each gene individually in various types of cancer, with GPSM2 so far being the most relevant to breast cancer.

GPSM2 is also known as LGN (Leu-Gly-Asn repeat-enriched protein), due to its N-terminal half containing a ten Leucine-Glycine-Asparagine repeats within to eight tetratricopeptide motifs [43]. This structure binds to NuMA, the nuclear-mitotic apparatus protein, and therefore establishes the important role of GPSM2 in modulating G protein signaling to mitotic spindle positioning during oriented cell division [44]. Furthermore, GPSM2 also plays a vital role in cell division symmetry control and in the G2/M phase of the cell cycle [45,46]. Cancer-related research has revealed that GPSM2 could potentially serve as a drug target and prognostic biomarker in hepatocellular carcinoma [47], lung adenocarcinoma [48], and pancreatic cancer [13]. GPSM2 was first reported to be highly over-expressed in breast cancer and play a crucial role in cytokinesis by Fukukawa et al. [8]. Subsequent studies confirmed this finding and also found that its nuclear expression is significantly linked to worse prognosis [11]. Interestingly, Zhang et al. demonstrated that GPSM2 is down-regulated in paclitaxel-resistant samples of breast cancer patients and could be a drug-resistance gene in breast cancer [10]. Consistent with previous research, we also found that the expression level of GPSM2 in breast cancer tissues increased significantly when compared to normal samples from Oncomine and UALCAN analysis. Survival analysis then showed breast cancer patients with high GPSM2 expression to have consistently worse distant metastasis-free survival. TIMER analysis, however, did not suggest a relationship between GPSM2 and immune infiltration patterns. Further experiments are required to validate these results. Taken together, we propose that GPSM2 is a potential prognostic marker and therapeutic target for breast cancer.

GPSM1 is also known as AGS3, as it encodes the activator of G protein signaling 3. GPSM1 gene has been found to be involved in cell division, reproduction, and differentiation [49,50,51]. Recent studies have identified GPSM1 as a predisposition gene of premature ovarian insufficiency [52], a plausible gene involved in regulation of skeletal muscle in diabetes mellitus [53], and to be overexpressed in prostate adenocarcinoma [12]. In our study, analyses from Oncomine and UALCAN suggested different results: GPSM1 was found to be under-expressed in breast cancer in the former database, while the latter indicated the reverse. This inconsistency could be due to the heterogeneity of datasets and nature of in silico analysis as a guide to further experimentally driven experiments, in vitro or in vivo [54]. Subsequent analysis did not show clear relationships between GPSM1 and survival time or immune infiltration levels in breast cancer.

GPSM3, alternatively known as AGS4 or G18, encodes for the activator of G-protein signaling 4, and was found to be overexpressed in breast cancer in our study. Previous research has indicated that GPSM3 is important in inflammation and autoimmune diseases, including rheumatoid arthritis and ankylosing spondylitis [55]. Furthermore, Robichaux et al. found that GPSM3 played critical role in the chemokine signal processing in leukocytes, and GPSM3-null mice exhibited significant neutropenia and leukocytosis [56]. These findings could explain why expression levels of GPSM3 were moderately to strongly correlated with all six types of immune infiltrate cells in breast cancer, using TIMER analysis. The immune system is indispensable in fighting cancer, and one of the most recent FDA-approved treatments for breast cancer includes anti-PD-1, an antibody-based immunotherapy that inhibits checkpoint on T cells [57,58]. However, only a subset of patients has been reported to see positive clinical response with such treatments. It is believed that understanding the cellular proportions, heterogeneity, and spatial distribution of the tumor immune microenvironment would help better stratify patients for whom immunotherapy is most beneficial [59]. While breast cancer was previously regarded as relatively non-immunogenic, it is now suggested that breast cancer is in fact rich in immune infiltrates, with varied functions and prognostic values [60]. In this study, we propose that GPSM3 plays a potential role in the diagnosis and inflammation in breast cancer.

GPSM4 is more commonly known as PCP2 (Purkinje cell protein 2), and is traditionally known to be highly expressed in cerebellar Purkinje cells and retinal bipolar neurons [61]. There are relatively few cancer-related reports about GPSM4. Reyes-Gibby et al. Reported that GPSM4 could be a novel target gene for neuropathic pain in patients with head and neck cancer [62]. In our study, Oncomine analysis showed that GPSM4 was differentially expressed in only breast cancer (downregulated) and prostate cancer (upregulated), among 20 types of cancer. Lower transcription levels of GPSM4 were found to be significantly related to worse distant metastasis-free survival outcome.

## 5. Conclusions

To summarize, our study proposed GPSM2 as a potential progsnotic marker and therapeutic target, and GPSM3 as a possible target for immunotherapy for breast cancer. Further in-depth experiments are warranted to confirm the role of these two GPSM members and their effectiveness as treatment for individuals with breast cancer.

## Figures and Tables

**Figure 1 biomedicines-09-01144-f001:**
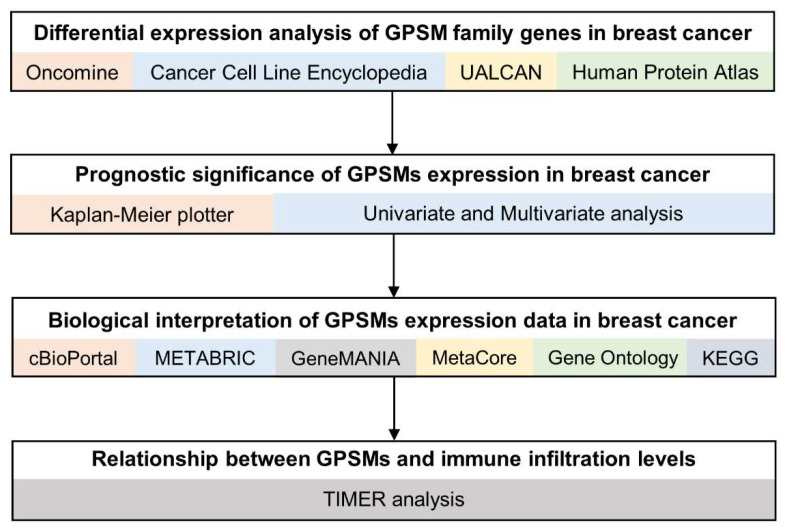
Study flowchart. METABRIC, Molecular Taxonomy of Breast Cancer International Consortium; KEGG, Kyoto Encyclopedia of Genes and Genomes; TIMER, Tumor Immune Estimation Resource.

**Figure 2 biomedicines-09-01144-f002:**
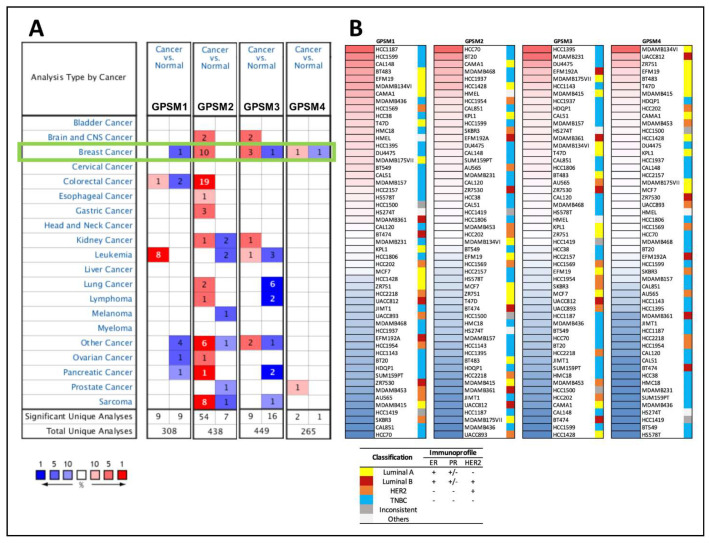
Expression of GPSMs across various cancer tissues. (**A**) mRNA expression levels of GPSMs in 20 cancer types. Numbers in red and blue cells represent dataset numbers in which levels of GPSMs are statistically increased or decreased, respectively (*p* < 0.05, fold change >2, gene rank top 10%, Oncomine). (**B**) Heatmap plots showing GPSMs expression status in breast cancer cell lines (CCLE), with colored columns on the right side displaying the molecular subtype of each cell line. “Inconsistent” denotes cell lines that are inconsistently annotated regarding the status of markers. “Others” include two cell lines that were not breast cancer (HMEL, engineered breast and HS274T, breast fibroblast). TNBC, triple negative breast cancer.

**Figure 3 biomedicines-09-01144-f003:**
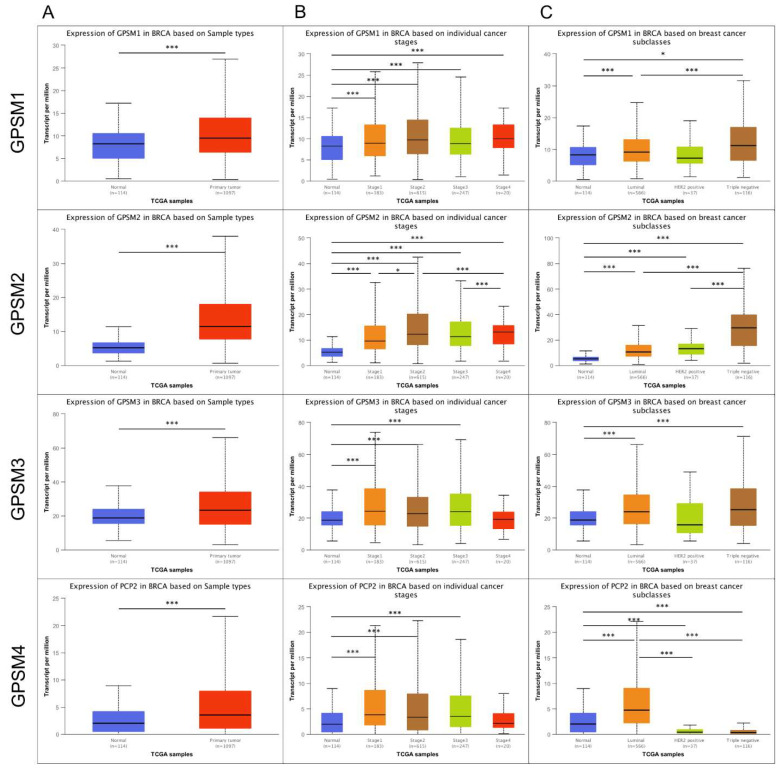
GPSMs expression in subgroups of people with breast cancer (UALCAN). Boxplots showing GPSMs transcript levels (**A**) in healthy controls versus individuals with breast cancer, (**B**) based on breast cancer stages, and (**C**) based on breast cancer subclasses. (* *p* < 0.05; *** *p* < 0.001).

**Figure 4 biomedicines-09-01144-f004:**
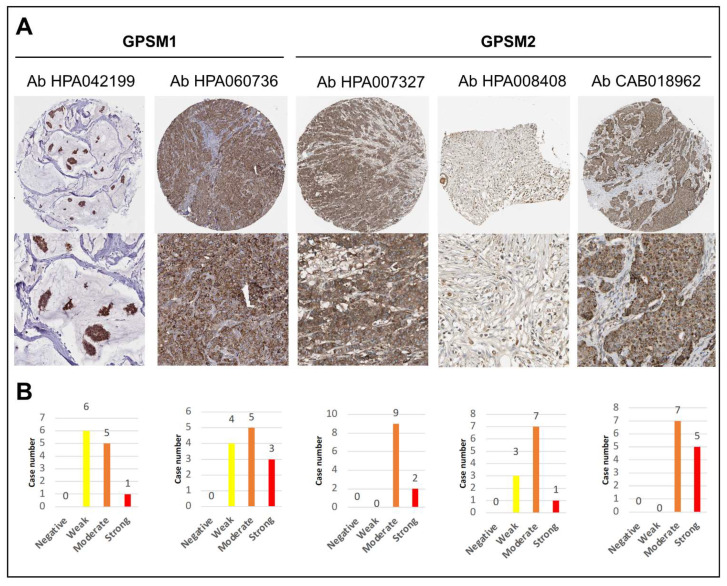
Protein expression profiles of GPSMs in breast cancer samples (Human Protein Atlas). (**A**) Representative strongly stained IHC images and (**B**) bar charts showing IHC staining intensity of GPSM1 and GPSM2 in breast cancer tissues. Corresponding data of GPSM3 and GPSM4 were not found. Ab, antibody; IHC, immunohistochemistry.

**Figure 5 biomedicines-09-01144-f005:**
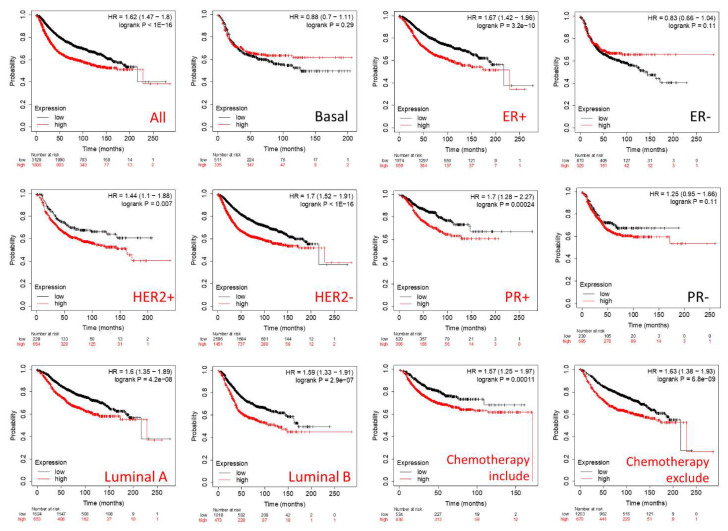
Distant metastasis-free survival (DMFS) analysis of GPSM2 in breast cancer (Kaplan–Meier plot). Red and black curves represent survival analysis for higher and lower GPSMs mRNA expression levels, respectively. Red and black titles indicate statistically and non-statistically significant survival, respectively. ER, estrogen receptor; HER2, human epidermal growth factor receptor 2; PR, progesterone receptor; +, positive; −, negative.

**Figure 6 biomedicines-09-01144-f006:**
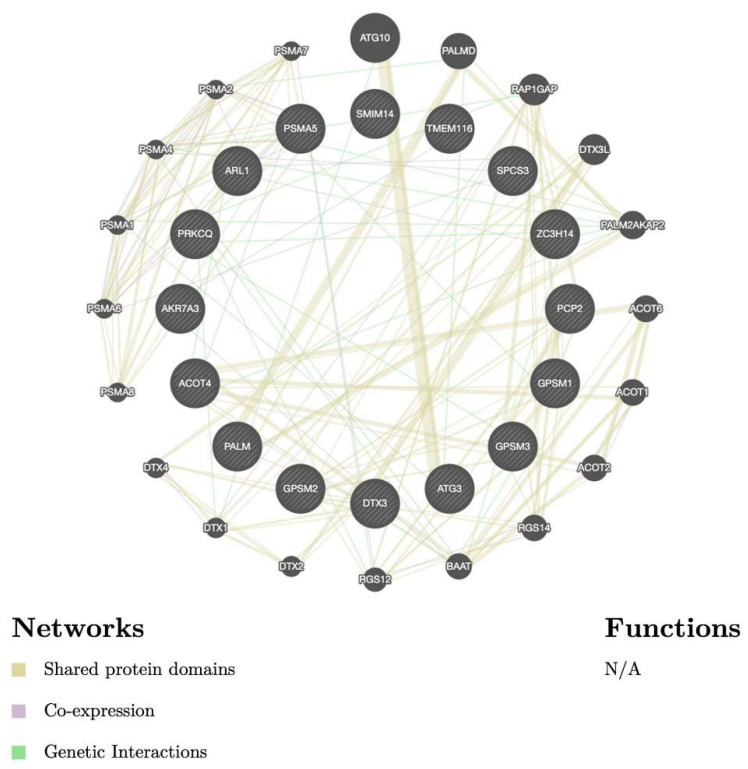
Gene–gene interaction network of GPSMs in breast cancer (GeneMANIA). Nodes represent genes, nodal sizes indicate interaction strengths, and line colors represent types of interactions.

**Figure 7 biomedicines-09-01144-f007:**
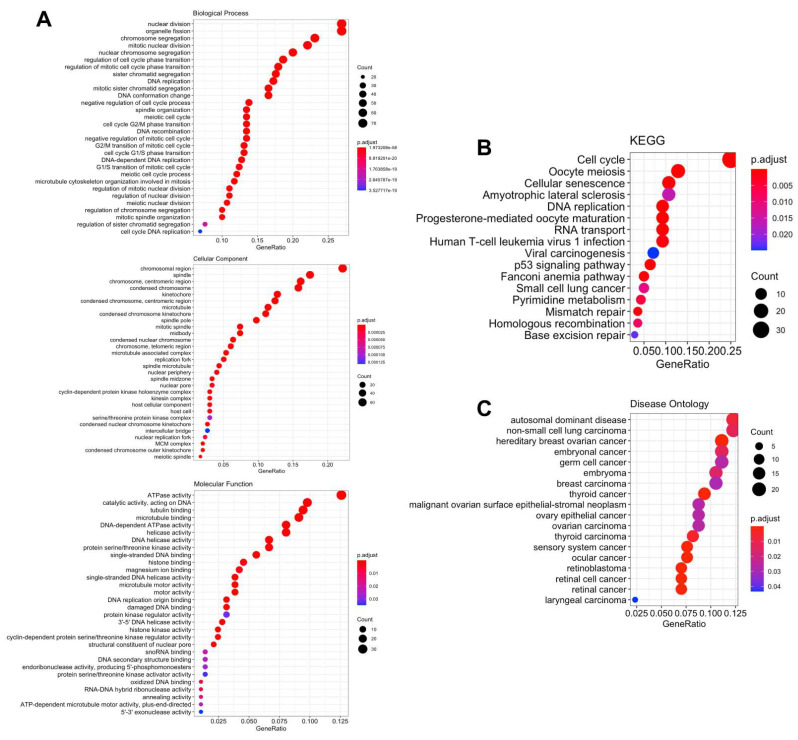
Functional and pathway enrichment analyses of GPSMs. (GO, KEGG). (**A**) Dot plot of Gene Ontology (GO) analysis. Shown are the most significant enriched categories for Biological process, Cellular component, and Molecular function. (**B**) Dot plot of Kyoto Encyclopedia of Genes and Genomes (KEGG) enriched analysis. (**C**) Dot plot of Disease Ontology analysis.

**Figure 8 biomedicines-09-01144-f008:**
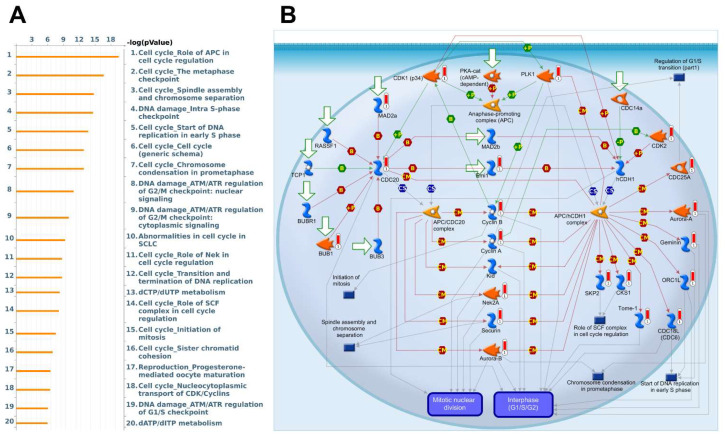
Enrichment pathway analysis of GPSM2 co-expressed genes in breast cancer database (MetaCore). (**A**) Pathway analysis. Potential gene networks and pathways affected by co-expressed genes (right column) and respective log *p* value (left column). (**B**) Biological process analysis. Symbols represent proteins. Arrows depict protein interactions (green, activation; red, inhibition). Thermometer-like histograms indicate microarray gene expression (blue, down-regulation; red, up-regulation).

**Figure 9 biomedicines-09-01144-f009:**
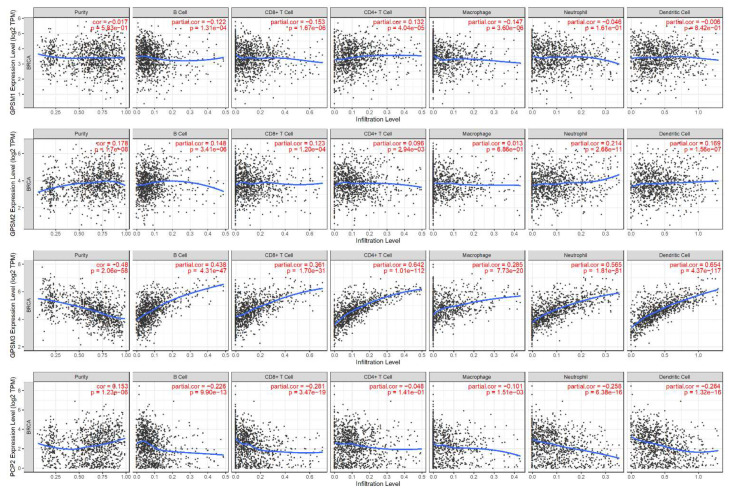
Relationships between GPSMs expression and immune infiltration level in breast cancer (TIMER). Horizontal axis, expression levels of GPSMs (values represented as log2 RSEM); vertical axis, tumor infiltrating immune cell markers (purity, B cell, CD8+ T cell, CD4+ T cell, macrophage, neutrophil, dendritic cell) TPM, transcript count per million; cor, correlation coefficient.

**Figure 10 biomedicines-09-01144-f010:**
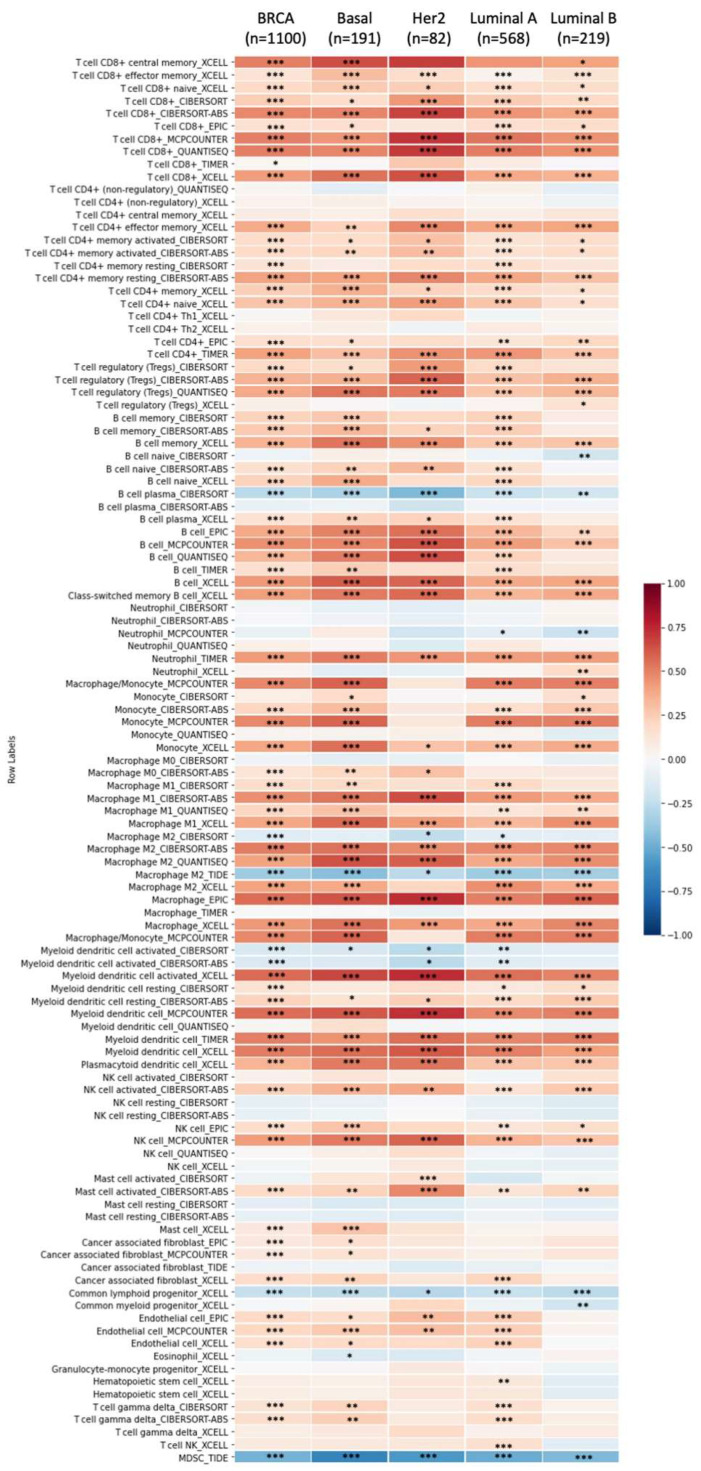
Relationship between GPSM3 expression and immune infiltration level in breast cancer (TIMER2). Correlation coefficients were calculated from seven cell type quantification algorithms (xCell, CIBERSORT, CIBERSORT abs. mode, EPIC, MCP-counter, TIMER, quanTIseq) and indicated as colors (blue, negatively correlated; red, positively correlated). * *p* < 0.05; ** *p* < 0.01; *** *p* < 0.001. Column: Subtypes of breast cancer, Row: Immune infiltrate levels.

**Table 1 biomedicines-09-01144-t001:** Basic characteristics of GPSM family genes.

Approved Symbol	HGNC ID	Gene ID	Aliases	Location on Chromosome
GPSM1	17858	26086	AGS3 DKFZP727I051	9q34.3
GPSM2	29501	29899	LGN Pins	1p13.3
GPSM3	13945	63940	NG1 G18 G18.1a G18.1b G18.2 AGS4	6p21.32
GPSM4	30209	126006	PCP2 MGC41903	19p13.2

**Table 2 biomedicines-09-01144-t002:** Univariate and multivariate Cox proportional hazards regression analysis of factors affecting overall survival. Variables with significant association with overall survival in univariate analysis were then put into the multivariate analysis. HR, hazard ratio; CI, confidence interval; *: significant risk factor, *p* < 0.05.

	Number of Patients	Univariate		Multivariate	
Variables		HR (95% CI)	*p* Value	HR (95% CI)	*p* Value
Age (y)					
<60	366	reference		reference	
≥60	287	2.032 (1.277–3.235)	0.003 *	2.361 (1.436–3.881)	0.0004 *
Gender					
Male	10	reference			
Female	643	0.887 (0.122–6.417)	0.906		
Tumor stage					
Stage I/II	471	reference		reference	
Stage III/IV	176	2.687 (1.657–4.356)	6.15 × 10^−05^ *	2.517 (1.516–4.18)	0.0002 *
M stage					
M0	530	reference		reference	
M1	9	3.965 (1.828–8.59)	0.0004 *	2.28 (1.014–5.128)	0.04 *
MX	114	1.268 (0.622–2.584)	0.513	0.726 (0.306–1.723)	0.49
T stage					
T1/T2	532	reference			
T3/T4	118	1.55 (0.920–2.609)	0.099		
GPSM2 expression					
Low	336	reference			
High	317	0.888 (0.56–1.41)	0.617		

## Data Availability

Not applicable.

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
