# Peer review of "Identifying GPSM Family Members as Potential Biomarkers in Breast Cancer: A Comprehensive Bioinformatics Analysis"

_biomedicines, 2021, doi:10.3390/biomedicines9091144_

Round 1

Reviewer 1 Report

The authors present the novelty data in case breast cancer therapy. 

I would like to know about the level of studied parameters in case healthy patients. If the Authors check the expression of GPSM in patient without breast cancer. ? 

Reviewer 2 Report

In this manuscript, Dong et al. illustrated the use of different bioinformatics tools to identify G protein signalling modulator (GPSM) as a possible biomarker for breast cancer. First, they compared the differential expression of GPSM1-4 across different cancer types and then breast cancer cell lines. Next, IHC results from the human protein atlas was employed to define the high, moderate and low expression in breast tumour tissue. Through KM analysis, they found GPSM2 would have a prognostic value in different subtypes of breast cancer. Subsequent pathway enrichment analysis discovered that GPSM2 should be related to cell cycle-related mechanisms and the correlation between GPSM1-4 and immune cell infiltration.

The authors illustrated a possible methodology for biomarker discovery in breast cancer, using GPSMs as examples. Before publication, the authors should address some issues shown below:

  1. In figure 1B, in addition to the expression of GPSM1-4, the authors should consider if the high expression of GPSM1-4 would correlate with a particular feature, e.g. ER+ve, HER2 overexpression or triple-negative. Therefore, the authors would demonstrate the possible clinical features that would associate with the high expression of GPSM1-4.
  2. In figure 2A, the authors should show all expression levels of GPSM1-4 instead of just low and moderate GPSM1 and GPSM2 expression. In figure 2B, the authors showed there was a strong staining result. Therefore, the database should have the IHC results.
  3. In figure 2, the authors could classify the protein expression of GPSM1-4. However, in figure 3, the authors performed the grouping based on mRNA expression. It seems the results from figure 2 were meaningless.
  4. In figure 5, the authors only showed the results for GPSM2. How about the other GPSMs? Also, there were A, B and C panels in the figure legend, but in figure 5, I cannot see these panels.
  5. In figures 7 and 8, the molecular function of GPSMs should relate to cell cycle regulation. However, in figure 9, the authors suddenly jumped to immune cell infiltration. What is the link between these figures?
  6. Finally, the authors suggested that GPSM could be a biomarker for breast cancer. In addition to overall survival, how about other clinical features, for example, disease-free survival, recurrence and metastasis? Apart from KM analysis, the authors should perform univariate and multivariate cox-regression analyses to determine the HR for each clinical feature.

Round 2

Reviewer 2 Report

The authors have already addressed all of my concerns. The manuscript is recommended for publication. 
